# Semantic priming supports infants' ability to learn names of unseen objects

**Elena Luchkina**[1]*ᵒ, **Sandra Waxman**[2,3]ᵒ

**1** Department of Psychology, Harvard University, Cambridge, Massachusetts, United States Of America,
**2** Institute of Policy Research, Northwestern University, Evanston, Illinois, United States Of America,
**3** Department of Psychology, Northwestern University, Evanston, Illinois, United States Of America

ᵒ These authors contributed equally to this work.
* elenaluchkina@fas.harvard.edu

**Data availability statement:** Data can be found in a public repository hosted by Open

## Abstract

Human language permits us to call to mind representations of objects, events, and ideas that we cannot witness directly, enabling us to learn about the world far beyond our immediate surroundings. When and how does this capacity emerge? To address this question, we evaluated infants at 12 and 15 months, asking whether they establish a representation of a novel noun's meaning in the absence of any visible referents, and use this representation to identify a candidate referent when it later becomes available. During training, infants (67 12-month-olds; 67 15-month-olds) were primed with words and images of objects from a particular semantic neighborhood (e.g., fruits) and were also introduced to a novel noun (e.g., "a modi"), used to name a hidden object. During test, infants heard that noun again, this time with two unfamiliar objects present—one from the primed neighborhood (e.g., a dragon fruit) and the other from an unrelated semantic neighborhood (e.g., an ottoman). If infants can represent something about the meaning of the novel noun in the absence of a visible referent and then use such a representation when a candidate referent appears, then at test, they should prefer the object from the primed semantic neighborhood. At 15 months, infants succeeded. In contrast, 12-month-olds did not succeed on this task even after a full week of vocabulary training designed to boost the effect of priming. It is possible then that 12-month-olds' representations of novel nouns' meaning are not yet sufficiently rich (if any at all) to guide their choice of referent when one does appear. Together, these findings suggest that the capacity to establish a representation of a novel noun's meaning in the absence of any visible referent and use this representation later to identify a candidate referent object emerges between 12 and 15 months.

## Introduction

Language is a powerful tool for learning and communication. It allows us to learn information that is not perceptually available at the time of learning [1], such as historic figures ('Napoleon'), hypothetical scenarios ('I might go out tomorrow'), or abstract scientific constructs ('subatomic force'). For toddlers hearing words for absent or unknown objects (e.g., "Mommy is outside fixing the trellis") and physicists conversing about quantum entanglement alike, *language enables conveying new information across space and time without requiring direct*

Science Framework: https://osf.io/g6bvd/
files/osfstorage?view_only=afa325c8a8c-
44c2aa63d819f14757d87. We include all
study data, except participants' race/ethnicity,
area of residence, family income, family size,
and primary caregiver education. These data
are considered personally identifying and are
not approved for public sharing by the IRB.
Primary caregiver education was include
in our analyses (we found non-significant
effects within preliminary analyses for both
Experiment 1 and Experiment 2). However,
requests for anonymized data can be transmit-
ted to the Northwestern University Institutional
Review Board via [sbsirb@northwestern.edu],
and anonymized data may be provided to eligi-
ble parties on reasonable request and with the
completion of any required prerequisites (such
as a Data Use Agreement).

**Funding:** This research was funded by an NIH
NRSA F32 Postdoctoral Fellowship awarded
to Dr. Elena Luchkina (GRANT13251277) by
Eunice Kennedy Shriver National Institute of
Child Health and Human Development. The
funders had no role in study design, data
collection and analysis, decision to publish, or
preparation of the manuscript.

**Competing interests:** The authors have
declared that no competing interests exist.

*perceptual access to the objects or phenomena in question.* Here, we investigate the developmental origins of this capacity. We ask whether infants can, in principle, represent something about the meaning of a novel noun (e.g., its semantic neighborhood, its plausible referents) in the absence of any visible referents and rely on those representations to identify those referents when they become visible. This is a crucial first step in the undoubtedly protracted developmental process of establishing, retrieving, and updating representations of phenomena based on language input alone.

Prior literature suggests that infants first begin to link *familiar* words to familiar perceptually unavailable items around their first birthday. For example, by 12 months, they comprehend "absent reference" – reference to objects that were previously seen but are currently hidden. When asked to locate a recently seen but now hidden object, 12-month-olds respond by pointing to its previous location, approaching that location, or looking towards it upon hearing its name [2–10]. By 14 months, infants use words to retrieve memories of past events [11] and properties of named, but currently absent, objects [12]. At first, around 12–14 months, infants rely on visual "anchors"– perceptually present reminders that scaffold their representation of the now-absent referent [13]. By 16 months, however, they comprehend absent reference in the absence of anchors [14], can update their existing representations based on language [15], and rely on category knowledge (e.g., the object is an apple) to interpret requests for unseen objects [16].

Although infants' comprehension of absent reference for familiar words emerges around 12 months, it is not until 19–24 months that they successfully *form a mental representation of the meaning of a novel word, introduced in absence of any visible referent* [17–21]. For example, Ferguson et al. (2014; [22]) presented 15- and 19-month-olds with novel nouns that were arguments of known verbs, either in animacy-selecting ("The dax is crying") or in animacy-neutral frames ("The dax is right here") in the absence of any visible referent. When the novel word was presented in animacy-selecting frames, 19-month-olds looked to an animate object when it later became visible. Fifteen-month-olds in the same paradigm did not show this preference. This pattern presents a puzzle. If 15- to 16-month-old infants can retrieve and update an existing mental representation of familiar word referents based on language alone, then why do they not infer the absent referent of a novel word when they have been provided with semantic cues?

One possibility is that when presented with semantic cues to meaning, but no visible referent of the novel word during the naming episode, 15-month-olds are not able to establish a mental representation, however sparse, of its possible referent due to representational constraints [23]. This seems unlikely; even younger infants can infer the presence of a hidden object when provided with linguistic and contextual cues [24]. Another possibility is that infants' relatively sparse vocabularies limit their success in using linguistic cues to infer the meaning of the novel word. That is, perhaps because 15-month-olds comprehend so few verbs ($M$=4.3; MacArthur Level II Short Form), they struggled to take advantage of the animacy-selecting known verb presented in Ferguson's task. Thus, it remains an open question when infants first succeed in mapping a novel word onto a representation that is sufficiently robust to permit them to identify a referent object when it later becomes available.

To address this question, we introduced a new experimental paradigm that (1) uses ostensive noun labeling to help infants infer the presence of an unseen object and (2) relies on semantic priming to convey something about candidate referents of that novel noun. In this paradigm we build on infants' relatively robust existing lexicon of nouns, their sensitivity to ostension [25–27], and their emerging sensitivity to semantic neighborhoods [28,29]. Prior literature documents that by 12 months, when shown images of two objects (e.g., "sock" and "juice") and prompted to look at an object not depicted (e.g., "foot"), infants look more to

the image that is semantically related to the prompt (e.g., "sock"; [30]). The literature also documents that by 12 months, infants successfully form object categories, especially in the context of naming events, and extend these names to novel exemplars of the same categories [30–33]. Grounded in this evidence, we leverage infants' conceptual capacities to invoke semantic neighborhoods.

Prior to conducting Experiments 1 and 2, we obtained pilot data from 18 infants in the Semantic Priming condition (described in detail in Experiment 1), using a Tobii X60 corneal reflection eyetracker. These infants revealed a significant looking preference for the target object at test, $M$=68.43%, $SD$=13.99%, $t(17)$=5.87, $p$ <.001, and a reliable increase in their looking to the target object each time the target novel word (one taught during training) was mentioned. These data provided assurances that the paradigm is well-suited for testing 15-month-olds' ability to represent something about the word's meaning, in the absence of a referent, and to use that representation when a candidate referent of the novel word appears.

## Deviations from stage-1 registered report

1. In the Stage-1 manuscript, the Switch Word condition was referred to as "Follow-up Control" condition. We chose a more informative label "Switch Word" condition for the Stage-2 submission.

2. The Stage-1 manuscript stated a planned cutoff of 15% of exposure to languages other than English. However, Lookit [34], which we used for data collection due to Covid-19 pandemic, discourages exclusions based on language exposure. As a result, we tested infants who varied widely in their exposure to languages other than English. Therefore, we relaxed the language exposure cutoff to 20%; this permitted us to keep the final sample size reasonably close to the planned one ($N$=72; determined by power analysis),

3. The Stage-1 manuscript stated that human coders would manually annotate infants' looking behavior if data were to be collected on Lookit. However, the advancement of automated gaze annotation technologies enabled us to save time and effort. Prior to implementing iCatcher+ [35] for gaze annotation, we estimated the agreement between iCatcher+ and human coders in our lab using an independent sample of $N$=48 12-month-olds. The average confidence of iCatcher+ was 95%. The agreement between iCatcher+ and human coders was 88%, excluding track loss (see Appendix 4). This excellent agreement, which aligns well with that reported by the developer team and with intercoder agreement for trained human coders on this experimental task in our lab (90%), was high enough to warrant adopting iCatcher+ for coding the full sample.

4. The originally planned window of analysis, 0–6000 ms, was based on Ferguson et al. (2014). We had initially planned to implement the same time window, but subsequently realized that the current design was closer to other word-learning tasks that involve ostensive labeling. To clarify, in Ferguson et al., to respond to the audio prompt, infants had to recall the sentence in which the name of the unfamiliar object had previously been embedded (e.g., "the dax is crying"), and use that sentence to infer the meaning of the novel noun. This incurred considerable processing time: differences between conditions in that design began to diverge around 3250 ms. In contrast, in the current design, as in most word-learning tasks used for this age, infants did not need to rely on sentence processing. Instead, they could infer the referent of the novel word from the semantic neighborhood within which it was presented. We therefore selected the window of analysis—367–2000 ms after the target word onset—that is used in most infant word learning designs (see [36]; we provide

additional citations in the text). This window is based on the time it typically takes infants to process a word and initiate a gaze shift. We analyzed looking preference for each of the two mentions of the target word, averaging across them to calculate the mean percent of looking to the target for each infant on each trial. This change does not affect the direction or the significance of our results. The effect size is smaller with the original window than with the new window.

5. The Stage-1 manuscript stated that the minimal amount of looking to the screen during test trials would be 16.67% of the entire duration of the trial (1 s out of 6 s of image display). However, here we report a more conservative cutoff of 50% (3 s out of 6 s of total image display). We did so to accommodate the new window of analysis—367–2000 ms from the onset of each mention of the target word. If we used 1 s as our cutoff and if that 1 s occurred between the two mentions of the target word, which were about 3 s apart, then no data would be available for analyses.

6. No Baseline assessment for Experiment 2 was included in the plan outlined in Stage-1 manuscript. However, we reasoned that this baseline measure was essential for evaluating the effect of vocabulary training.

7. In the Stage-1 manuscript, we stated that the plan was to ask caregivers for 14 days of picture book reading. This requirement, however, proved too demanding; Lookit requested that we shorten the vocabulary training to 7 days. In addition, to ease caregivers' burden, we did not ask them to report their infants' engagement during each reading session (also planned in Stage-1 manuscript).

## Experiment 1

There is ample evidence that 15-month-olds successfully retrieve representations of recently seen hidden objects upon hearing their names [37,38]. Yet, to our knowledge, there is no evidence that they build a representation for the meaning of a novel word, with no referent in sight, and use that representation to identify a candidate referent when one becomes visible. Experiment 1 was designed to test this capacity in 15-month-old infants.

### Method

**Participants.** Participants were 67 infants (*N*=23 in the Semantic Priming, *N*=23 in the Switch Word, and *N*=21 in the No Priming condition) recruited on Lookit, an online platform for developmental research. We selected Lookit because it permitted us to collect data during the Covid-19 pandemic. Thirty-one additional participants were excluded from the final sample due to technical failure (*N*=10), not meeting study criteria (e.g., more than 20% of exposure to languages other than English; *N*=17), missing MCDI or language exposure data (*N*=3), and data withdrawal (*N*=1). Caregivers completed a Qualtrics survey containing a MacArthur Short Form Vocabulary Checklist: Level II, Form A [39], augmented with words used during familiarization and with questions about infants' exposure to languages other than English (see Appendix 1). Caregivers also completed a standard Lookit demographic survey (see Appendix 2).

All participants, regardless of their inclusion in the final sample, received a $5 Amazon. com gift card. Survey information for the final sample is summarized in Table 1.

**Timeline.** Data for Experiment 1 were collected between 17/11/2020 and 13/11/2021

**Apparatus.** Caregivers logged into Lookit on their desktop or laptop computers to access the study and completed a consent process by video-recording their verbal consent with a web

**Table 1. Demographic and vocabulary survey summary, Experiments 1 and 2.**

| Measure | | Experiment 1 | Experiment 2 |
|---|---|---|---|
| | | M (SD) | M(SD) |
| Number of known nouns (per MCDI-I) | | 23.7 (15.2) | 11.4 (12.3) |
| Number of known verbs (per MCDI-I) | | 3.2 (2.8) | 1.6 (2.2) |
| Number of known adjectives (per MCDI-I) | | 0.8 (1.8) | 0.3 (1.1) |
| Total number of known words (per MCDI-I) | | 34.9 (22.5) | 17.5 (18.1) |
| Exposure to other languages, % | | 2% (5%) | 1% (4%) |
| Age, days (in Exp. 2 – before vocabulary training) | | 446 (19.6) | 359.9 (22.6) |
| Age, days after vocabulary training (Exp. 2 only) | | | 373.7 (22.6) |
| Gestational age, weeks | | 38.8 (1.6) | 39.0 (1.0) |
| Number of children in the household | | 1.8 (1) | 1.4 (0.7) |
| Number or caregivers in the household | | 2 (0.2) | 2.0 (.02) |
| **Measure** | | N | N |
| Gender | Male | 42 | 32 |
| | Female | 27 | 37 |
| Ethnicity | Asian | 4 | 4 |
| | Black or African American | 0 | 1 |
| | White | 45 | 49 |
| | More than one ethnicity | 19 | 15 |
| Primary caregiver's age | <18 | 1 | 2 |
| | 20-29 | 9 | 16 |
| | 30-39 | 48 | 48 |
| | 40-49 | 10 | 3 |
| Primary caregivers' education | High School | 5 | 3 |
| | Some college | 0 | 1 |
| | Bachelor's degree | 15 | 14 |
| | Graduate degree | 1 | 1 |
| | Professional degree | 47 | 50 |
| Household income, $K | 15-60 | 12 | 11 |
| | 70-100 | 12 | 14 |
| | 110-150 | 15 | 20 |
| | >160 | 21 | 20 |
| | Not provided | 8 | 4 |

camera. Caregivers were then asked to hold their infants about 50–60 cm from their monitor, positioning them over the shoulder, to ensure that only the infants' (and not their own) eye movements would be recorded via the web camera. First, infants participated in a calibration phase during which a colorful spinning abstract shape moved around the monitor screen, with each movement accompanied by a chime sound. Then the experiment proper began. All stimuli were presented on caregiver's desktop or laptop computers. Infants' gaze during the study was recorded with a web camera, similar to how it would be recorded in a lab by a camera located underneath or above the screen displaying experimental stimuli. Most videos were recorded at 30 HZ per second; all were resampled to ensure the same rate of 30 Hz per second. Video recordings were stored on Lookit's servers.

**Stimuli.** Infants viewed four video-taped vignettes, each featuring a novel word-object pairing. Each video began with an actor pointing to three familiar objects, which appeared behind her, but were visible to the infant. She named each familiar object with its basic-level count noun (e.g., "apple", "orange", "banana"). Next, the actor turned her head to look toward another object"), also behind her but this time not visible to the infant (it was obscured by her body), and named it (e.g., "modi; see Fig 1 for a representation of a single trial and Appendix

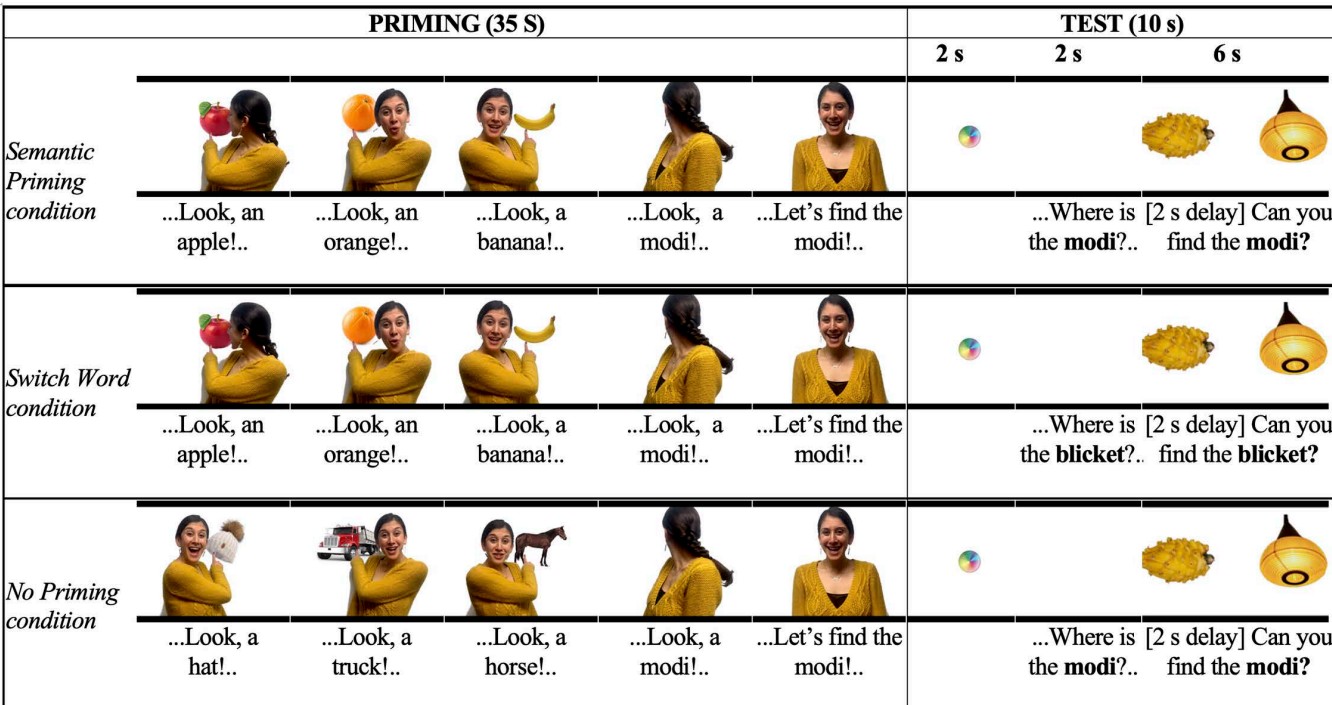

**Fig 1. A representative example visual\* and auditory information presented during the Priming and Test phases in the Semantic Priming, Switch Word, and No Priming conditions.** Pointing and ostension cues were provided to support an inference that the last episode of the Priming phase, in which the object is hidden, was a labeling episode as well. \*Source of images in depicted here: https://unsplash.com. Actual stimuli may differ (all stimuli are available on OSF).

3 for a complete list of stimuli). All stimuli are available on the Open Science Framework: https://osf.io/g6bvd/?view_only=afa325c8a8c44c2aa63d819f14757d87.

**Stimulus selection and design.** We selected visual and linguistic stimuli based on vocabulary norms [40], focusing on the four semantic neighborhoods—fruits, clothing, vehicles, and animals – that include nouns that most 15-month-olds (at least 60%) comprehend. In the Priming phase, infants viewed images of familiar objects from these semantic neighborhoods: apple, banana, orange (fruits), jacket, sock, hat (clothing), truck, car, bus (vehicles), cat, dog, horse (animals). In the test phase, infants viewed images of two unfamiliar objects for which most 15-month-olds do not have a name. One (the target object) was a member of the same semantic neighborhood as objects presented during Priming. These included a Gursky's spectral tarsier (animals), a doctoral gown (clothing), a drone car (vehicles), and a dragon fruit (fruits). The distractor object was an unfamiliar member of a different semantic neighborhood, unrelated to objects presented during Priming, including a coil of rope (tools), a spatula (utensils), a primula coffee maker (vessels), and an ottoman (furniture). All images were approximately the same in size, resolution, and their levels of contrast and saturation.

**Procedure.** Experimental sessions were conducted at participants' homes at the time of their choosing. The procedure, conducted entirely on Lookit, was 'asynchronous'—with no experimenter present. Before the task began, caregivers viewed instructions displayed on the screen and provided verbal consent; their consent was video recorded and then reviewed by trained research assistants.

Each infant completed 4 trials (within-subject), corresponding to the four semantic neighborhoods, each including a Priming and a Test phase (Fig 1). Infants were assigned randomly

to conditions (between subjects). The target image appeared on alternating sides across four trials (e.g., left-right-left-right or right-left-right-left). We created 4 possible sequences of trials using a Latin square design. Infants were assigned randomly to one of these sequences. We implemented a between-subjects design to avoid potential carry-over effects, which could dilute the predicted advantage of semantic priming. In the Semantic Priming and Switch Word conditions, all Priming phase objects were members of the same semantic neighborhood; what varied by condition was the word presented at test. In the No Priming condition, priming objects were members of different semantic neighborhoods; the test phase was identical to that in the Semantic Priming condition.

Semantic priming condition: Priming (duration: 34–37 s; $M_{across\ trials}$=35.5 s): An actor pointed to and named three familiar objects, all from the same semantic neighborhood. These appeared one at a time, behind an actor who turned to point to each object as it appeared and name it with its familiar basic-level category name (e.g., "Look! An apple! Do you see the apple?"). The fourth object was entirely occluded by the actor's body (seemingly unintentionally) and thus hidden from infants' view. As in the first three trials, the actor turned, pointed, and produced a novel noun (e.g., "Look, a *modi*!"). The actor alternated her gaze between the camera (to emulate eye contact with the infant) and the space behind her, to indicate that the referent of the novel noun was behind her.

Test (duration: 10 s): Infants first saw an attention getter (2 s), followed by a blank white screen (2 s), during which they were prompted to look to the object corresponding to the novel label, e.g., "Now look! Where is the modi?". Next, two unfamiliar objects appeared side-by-side—one a member of the primed semantic neighborhood and the other a member of a different semantic neighborhood. With the test objects visible, infants heard a second verbal prompt, e.g., "Can you find the modi?".

Switch word condition: The procedure was identical to the Semantic Priming condition with one crucial exception: during Test, infants heard a different novel noun than the one introduced in Priming. This condition was designed to assess whether infants prefer the target object based on semantic priming alone, independent of the novel noun presented during Priming to name the hidden referent.

No priming condition: The procedure was identical to the Semantic Priming condition with one exception: during Priming, the three familiar word-object pairs were each selected from different semantic neighborhoods (e.g., a hat, truck, and horse) and unrelated to the test objects. The control condition was designed to assess side biases or intrinsic preferences among the test pairs.

We designed the Switch Word and No Priming conditions to maximize their similarity to the Semantic Priming condition. This was done to isolate the effects of our manipulation from any asymmetries in exposure to audio and visual information or differences in attentional depletion. For example, in the Switch Word condition, presenting infants with test images in silence instead of using an unfamiliar novel noun could also assess whether semantic priming alone explains infants' looking preference. In the No Priming condition, instead of labeling three semantically distant objects, we could have omitted the priming phase entirely. However, the patterns of looking across conditions would have been incomparable because infants in the Semantic Priming condition would have been exposed to more information and required to distribute their attention differently than in the other two conditions.

**Data preparation.** Infants' eye-gaze was coded frame-by-frame (1 frame=33.33 ms) by iCatcher+ software. We included only those trials on which infants looked to the screen at least 80% during training and at least 50% of the time during test trials. This yielded 158 trials with each participant contributing on average 2.9 trials out 4. For each trial and each infant, we calculated two dependent measures:

1. **Mean %LT to the target**. We calculated the mean percent of looking time devoted to the target [looking to target object/(looking to target + distractor)] during the window of analysis. That window was the average of the percent of looking time devoted to the target object across two segments 367–2000 ms after the target word onset—for each of two mentions of the target word. Hereafter, "target" and "target object" refers to the object defined as the target one in the Semantic Priming condition. Although the same pairs of objects were used in test trials in all three conditions, in the Switch Word and No Priming conditions neither object could truly be considered a target.

2. **Time course**. We aggregated looking data into 200-ms bins throughout the duration of image display and calculated the percent looking to the target during each bin. This permitted us to assess the time course of infants' looking to the target and the distractor and identify divergencies among looking trajectories across different conditions. Cluster-based permutation tests were conducted with 1000 samples for each 200-ms bin.

**Planned analyses (based on Stage-1 manuscript).** We assessed the effect of Condition on the mean %LT to the target and the time course of infants' %LT for each 200-ms bin.

1. Mean %LT to the target We used a Generalized Linear Mixed Model (GLMM) with Condition as a fixed factor and Test Item (fruits, clothing, vehicles, or animals) and Participant as random factors (grouping variables) for the intercept. This helped us reduce Type-1 error by including a grouping variable that prevented inflation of the degrees of freedom [41] and accounted for any differences in infants' knowledge across the four semantic neighborhoods.

For each model, we tested the residuals for normality using a Shapiro-Wilk test to ensure accurate inferences based on p-values in estimating the model coefficients. Unless explicitly stated otherwise, the distribution of residuals of our models did not deviate significantly from normal. Unless otherwise specified, Semantic Priming was the reference level for Condition (to contrast it with the other conditions).

2. Time course. We used a cluster-based permutation analysis [42] to identify any significant clusters of divergence among conditions following the mention of the target word. The threshold was established by a *t*-value of 1.69 for each bin, which corresponds to the alpha=.05 for an independent-sample one-tailed t-test with 36 degrees of freedom (the number of participants per group was lower than the final sample due to additional exclusions—see above).

The effects of demographic factors—age, gender, and mother's education—vocabulary scores, and trial order were tested in a preliminary analysis, implemented by fitting a GLM to the DVs. An additional set of preliminary analyses evaluated the effects of infants' knowledge of the words used in priming (based on caregivers' vocabulary survey). Trial order was included to test whether infants' looking preferences change over the duration of the procedure. In subsequent analyses, we included only those factors that yielded significant effects in the preliminary analysis. Doing so reduces the chance of multicollinearity (which increases with the number of multilevel categorical predictors included) or an overly specified model (that cannot be supported by the data given our limited sample size *N*=67).

**Predictions.** By 15 months, infants infer the presence of a hidden object by following a speaker's line of regard [43], understand that words communicate about mental states [44–46], appreciate referential cues [47], and leverage category membership to extend the meanings of novel nouns. Based on this evidence, we made the following predictions:

**1. Mean % LT to the target.** Semantic priming condition: We predicted that the mean %LT to the target would be significantly greater than chance (50%) and higher than in the Switch Word and No Priming conditions.

Switch word condition: If infants' looking in the Semantic Priming condition was driven by semantic priming only, unrelated to the novel noun, then the mean %LT in the Switch

Word condition should not differ from that in the Semantic Priming condition. Conversely, if infants' looking in the Semantic Priming condition reflected the representation they had formed for the meaning of the novel noun presented in Priming then their mean %LT in the Switch Word condition should be significantly lower and not different from chance. Although the principle of mutual exclusivity might predict that infants in the Switch condition would interpret the new word as referring to the distractor object, we note here that evidence concerning mutual-exclusivity inferences in infants younger than 18 months is mixed (e.g., [48] vs. [49]).

No priming condition: If infants have no intrinsic preferences for either of the test objects, then their mean %LT to the target should not deviate from chance.

**2. Time course.** We predicted that in the Semantic Priming condition, if infants' looking reflected their inferences about candidate meanings of the novel noun, they would increase their %LT to the target object after each mention of the target word (as observed in the pilot). We also predicted that their looking trajectory would significantly diverge from the looking trajectories in the Switch Word and No Priming conditions. Alternatively, if infants' looking in the Semantic Priming condition was driven by semantic priming alone and was unrelated to the novel noun, their looking would be unrelated to the verbal prompts and they would prefer the target object consistently throughout the trial. In this latter case, we would expect the same pattern of results in the Switch Word condition and no significant divergencies in looking trajectories between the Switch Word and Semantic Priming conditions. We predicted that in the No Priming condition, the %LT to the target would not be affected by the verbal prompts and remain close the chance level throughout the trial. If we observe any consistent looking to either test object in the No Priming condition, it would suggest intrinsic looking preferences.

## Results and discussion

The results provide new evidence that 15-month-olds can build a representation of the meaning of a novel noun while its referent is hidden and can subsequently draw on this representation to identify a candidate referent when it becomes visible.

## Results of planned analyses

**1. Mean % LT to the target.** First, we analyzed infants' visual attention to the screen at test. Infants contributed approximately equivalent amount of looking data on each of four test trials with no significant decline in their visual attention over the duration of the experimental procedure. Preliminary analyses, conducted for each condition independently (with all predictors entered at once), revealed no significant effects of primary caregivers' education level, or infants' age, gender, comprehension of familiar words introduced in priming, vocabulary (MCDI), or trial order on the mean %LT to the target object in any condition. These demographic and vocabulary measures were not included in subsequent analyses.

To evaluate the effect of Condition, we fitted a GLMM to the mean %LT. We offset the mean %LT by 50% (chance level) to improve interpretability of the effect of the intercept ($\beta_0$, the grand mean when all predictors are set to 0). The effect of the intercept was significant, $\beta=0.18$, $SE=0.05$, $t=4.0$, $p=.003$, indicating that infants in the Semantic Priming condition (which was the reference condition) preferred to look to the target. Moreover, their mean %LT to target was significantly higher in the Semantic Priming condition than in either the Switch Word condition, $\beta=-0.12$, $SE=0.05$, $t=-2.5$, $p=.01$, and No Priming condition, $\beta=-0.18$, $SE=0.05$, $t=-3.4$, $p=.001$ (Table 2).

**Table 2. Results of mixed-effect linear model fitting with Condition as a fixed factor and Test Item and Participant as random factors for the intercept.**

|  | β | SE | T-value | p-value |
|---|---|---|---|---|
| (Intercept) | 0.18 | 0.05 | 4.00 | .003* |
| Condition = No Priming | -0.18 | 0.05 | -3.4 | .001* |
| Condition= Switch Word | -0.12 | 0.05 | -2.5 | .01* |

An additional GLMM with No Priming as a reference level for Condition showed no significant difference between the Switch Word and No Priming conditions, β=0.06, SE=0.05, t=1.23, p=.22. The effect of the intercept was also not significant, β=-0.01, SE=0.04, t=-0.11, p=.91, which indicates that infants had no looking preference in this condition. A GLMM with Switch Word as the reference level for condition revealed no significant effect of the intercept either, β=0.06, SE=0.04, t=-1.76, p=.21, which indicates no looking preference in the Switch Word condition either (see Fig 2).

**2. Time course.** As predicted, infants' looking trajectories in the Semantic Priming and Switch Word conditions significantly diverged after the first mention of the target word, at 200–1200 ms, sum t-statistic=14.263, p=.026. Infants' looking trajectories in the Semantic Priming and No Priming conditions significantly diverged after the second mention of the target word, at 3400–4200 ms, sum t-statistic=9.74, p=.04 (Fig 3).

In the Semantic Priming condition (M=68.22%), the experimental data aligned with pilot data (M=68.43%) for this condition. Time course analyses revealed no significant clusters of divergence among looking trajectories obtained from Lookit and a Tobii automated eyetracker

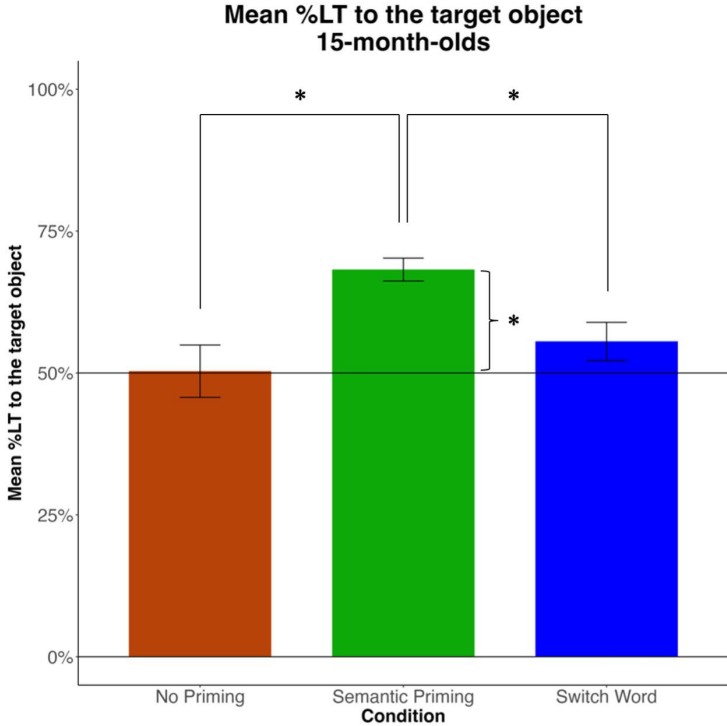

**Fig 2. Mean %LT to the target across two windows of analysis (after each mention of the target word), by Condition.**

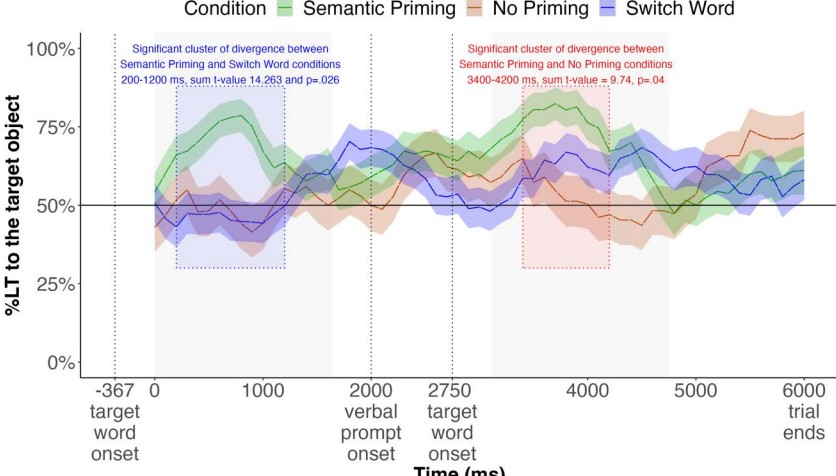

**Fig 3. Time course of the %LT over the course of the trial for infants in each Condition.** Windows of analysis (367-2000 ms after each target word onset) are shaded grey. The onset of image display coincides with the offset of the target word.

([Fig 4]). This provides assurances of replicability of results obtained using an eyetracker, and online, using participants' web cameras.

**Discussion.** We reasoned that if 15-month-olds could establish a representation, however sparse, for the meaning of the novel noun in the absence of a referent and use it to identify a candidate referent later, when one becomes visible, then infants in the Semantic Priming condition (but in neither of the control conditions) would prefer to look to the target test

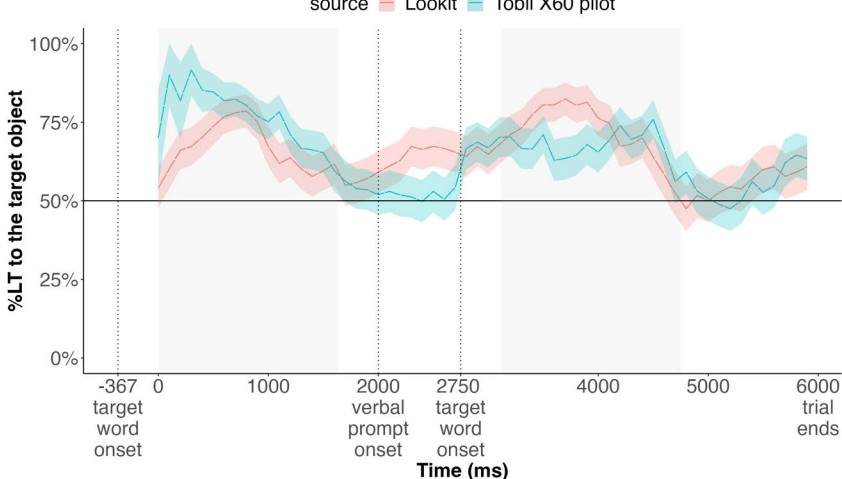

**Fig 4. Time course of infants' looking during the trial in the Semantic Priming Condition obtained in lab (Tobii X60) and on Lookit.** Windows of analysis are shaded grey.

object. Our results are consistent with this prediction. The absence of looking preferences in either of the two control conditions—No Priming and Switch Word—suggests that infants' success in the Semantic Priming condition cannot be attributed to semantic priming alone or to any intrinsic preferences or processing biases (e.g., object size, animacy; see [50,51]). Instead, their success reveals an ability to represent something about meaning of a novel word while the referent is hidden and subsequently draw on this link and recognize that referent when it becomes visible.

## Experiment 2

The goal of Experiment 2 was to assess whether 12-month-olds, like 15-month-olds, can succeed on our task. There is reason to suspect that 12-month-olds' comprehension of absent reference, coupled with their capacity to link words to objects and object categories [52], may succeed on this task. Notice, however, that for the semantic priming manipulation to work, infants must comprehend the words introduced during priming for each semantic neighborhood. This presents a challenge: on average, only 30% of 12-month-olds meet this requirement [53]. We therefore introduced a 7-day vocabulary training period (described below), designed to bolster 12-month-olds' vocabularies, and measured their performance on experimental task of Experiment 1 both before and after vocabulary training.

### Method

**Participants.** Participants were 67 infants (*N*=22 in the Semantic Priming, *N*=25 in the Switch Word, *N*=20 in the No Priming condition) recruited on Lookit. An additional 124 infants were also recruited but excluded from the final sample due to (1) caregivers' failure to complete the full protocol, including all 9 vocabulary learning sessions (*N*=109), (2) > 20% of exposure to languages other than English (*N*=11), and missing survey data (*N*=4). All participants, regardless of their inclusion in the final sample, received a $5 Amazon.com gift card after completing the first vocabulary training session and an additional $10 Amazon.com gift card after the final session of the study if they completed it. Demographic and vocabulary survey information for the final sample is summarized in Table 1.

Apparatus, stimuli, and data preparation, IVs, DVs, and planned analyses were identical to those in Experiment 1, with a single exception: we added a new DV: the percent of looking to the target on the word knowledge test that we conducted at the completion of the 7-day vocabulary training period. As in Experiment 1, we included trials on which infants looked to the screen at least 80% during training and at least 50% of the time during test trials. This yielded 161 trials with each participant contributing 3.1 trials at the baseline assessment (before vocabulary training) and 152 trials with each participant contributing on average 2.9 trials at the outcome assessment (after vocabulary training).

**Timeline.** Data were collected between 5/10/2022 and 22/12/2023

**Procedure.** The experimental procedure included 4 parts, conducted over 9 sessions, all on Lookit:

1. **Baseline assessment** (identical to Experiment 1). This baseline assessment, when compared with the post vocabulary-training outcome assessment, enables us to evaluate whether and how performance on the experimental task was affected by vocabulary training.

2. **Vocabulary training** (7 days). We introduced a vocabulary training period designed to bolster infants' comprehension of the basic-level names of the objects introduced in the Priming phase. We presented caregivers with a digital picture book, asking them to 'read' it aloud to their infants once a day for 7 consecutive days, pointing to each image and

naming it with its familiar basic-level noun. If a caregiver missed a day of reading, we contacted them to remind them to continue their participation (about half of the final sample received such a reminder). To be included in the final sample, infants had to have participated in all 7 days of training.

**Word-image pairs in the picture book.** Twenty-four word-image pairs were presented in a random order. Twelve pairs were those used in priming, three from each of the four semantic neighborhoods—fruits (orange, apple, banana), clothing (jacket, sock, hat), vehicles (car, bus, truck) and animals (cat, dog, horse). The remaining 12 word-image pairs were drawn from the semantic neighborhoods of the distractor test objects —furniture (sofa, chair, table), tools (saw, wrench, hammer), utensils (fork, spoon, knife), and vessels (cup, glass, bowl). During vocabulary training, infants were not exposed to any of the test images used the main experimental procedure.

3. **Vocabulary test**. After vocabulary training was completed, we tested infants' comprehension of the words taught during Priming. To do so, we paired each of the 12 word-image pairs used in priming (cat, dog, horse, jacket, sock, hat, orange, apple, banana, car, bus, truck) randomly with one of the 12 word-image pairs drawn from the semantic neighborhood of the distractor objects (cup, glass, bowl, fork, spoon, knife, saw, wrench, hammer, sofa, chair, table). The side on which the target image appeared was determined randomly on each trial. Each trial lasted 10 seconds: for 2 seconds, the image appeared in silence, followed by a verbal prompt: "Look, a [word]! Where is the [word]?" Verbal prompts included the names of the 12 objects used in priming. We did not assess comprehension of the names for the 12 distractors because they did not figure in Priming.

4. **Outcome assessment** (identical to Experiment 1). Finally, infants' performance on the main experimental task was re-assessed. To evaluate the effect of vocabulary training, the stimuli for the baseline and outcome assessments were identical.

**Predictions.** For the *Baseline Assessment*, we predicted no differences among conditions on both the mean %LT and time course of the %LT. This outcome would be consistent with the possibility that infants have the requisite capacity to represent unseen object and to learn their names, but do not have a sufficient lexicon to support inferences about that object based on semantic priming.

For the *Vocabulary Test*, we predicted that infants would comprehend the words used in Priming in the main experimental procedure, yielding an average of about 70% looking to the target image.

For the *Outcome Assessment*, we predicted that infants' mean %LT to the target in the Semantic Priming condition would significantly exceed chance responding (50%) and mean %LT to the target in Switch Word and No Priming condition. We also expected that infants' time course for the %LT in the Semantic Priming condition would significantly diverge from both Switch Word and No Priming conditions, mirroring the results of Experiment 1. Finally, we predicted that in the Semantic Priming condition, the mean %LT at the outcome assessment would be higher than the mean %LT at the baseline assessment and that this difference between the baseline and outcome assessments would be correlated with infants' performance on the vocabulary test.

## Results and discussion

We assessed whether 12-month-olds could establish a representation for the meaning of novel words, with no referent in sight, and use that representation to identify a candidate referent when one becomes visible. We assessed this before and after a 7-day-long vocabulary training

provided by caregivers to boost 12-month-olds' comprehension of the familiar nouns used in Priming.

## Results of planned analyses

**1. Mean % LT to the target.** Preliminary analyses evaluated the effects of participants' age and gender, primary caregivers' education, and trial order. As in Experiment 1, these analyses were conducted independently for each condition and for each time of testing (baseline and outcome), using the mean %LT to the target as the DV. All predictors were entered into the model at once. Only one effect was significant: the effect of Age in the Semantic Priming condition *at the outcome assessment*, $\beta$=-0.005, $SE$=0.002, $t$=-2.53, $p$=.015. We suspect that this unpredicted effect is the result of a Type-1 error. Nevertheless, we included an Age-by-Condition interaction in our main analyses for both baseline and outcome assessments. An additional set of models tested the effect of infants' comprehension of the 12 words used during Priming prior to the vocabulary training (the DV was derived from the MCDI). This analysis revealed no significant effects for either condition at baseline and outcome assessments. This variable was not included in subsequent analyses.

**Baseline assessment (before vocabulary training).** A GLMM fitted to the mean %LT to the target evaluated the effects of Condition, Age (in days), and Age-by-Condition interaction. As in Experiment 1, we offset the mean %LT during Test by 50% to improve interpretability of the effect of the intercept. Only the effect of the intercept ($\beta_0$) was significant, $\beta$=0.14, $SE$=0.05, $t$=2.94, $p$=.022, indicating that infants' looking in the Semantic Priming condition was significantly above the chance level. The distribution of residuals, however, significantly deviated from normal ($p$<.05) and was left-skewed, rendering significance levels uninterpretable. To address this issue, we applied an exponential transformation and re-ran the analyses. The results (Table 3) did not differ substantially from the original model, except that residuals were normally distributed.

Next, to compare infants' performance in the two control conditions to the chance level and to one another, we used two additional models—one with No Priming and the other with Switch Word as the reference levels for Condition. The effects of the intercept were significant in both models: $\beta$=0.45, $SE$=0.05, $t$=8.37, $p$<.001 and $\beta$=0.43, $SE$=0.05, $t$=8.33, $p$<.001 for No Priming and Switch Word conditions treated as the reference level, respectively. This reveals that infants preferred the target object in both control conditions; there were no significant differences between these conditions, $\beta$=-0.02, $SE$=0.05, $t$=-0.44, $p$=.66.

**Vocabulary training.** A visual inspection of the vocabulary training videos confirmed that caregivers named all images in the picture book. We then measured the mean %LT to the target during two windows of analysis (367–2000 ms from the target word onset), corresponding to the two mentions of the target word. We used a one-sample t-test to compare infants'

**Table 3. Results of mixed-effect model fitting with Condition as a fixed factor and Test Item and Participant as random factors for the intercept. Baseline assessment.**

| | β | SE | T-value | p-value |
|---|---|---|---|---|
| (Intercept) | 0.45 | 0.05 | 8.32 | <.001* |
| Condition=No Priming | 0.00 | 0.05 | 0.02 | .98 |
| Condition=Switch Word | -0.02 | 0.05 | -0.42 | .68 |
| Age in days at Baseline, centered at median | 0.00 | 0.00 | -1.97 | .05 |
| Condition=No Priming*Age in days at Baseline | 0.00 | 0.00 | 1.10 | .27 |
| Condition=Switch Word*Age in days at Baseline | 0.00 | 0.00 | 1.08 | .28 |

knowledge of each word to chance. Infants' looking to the target was significantly above chance (all *p*-values<.02) for 9 out of 12 items (Fig 5). Infants' looking did not significantly differ from chance on the 3 remaining items—'sock,' 'banana,' and 'jacket' trials. There was no significant correlation between word comprehension (reported by caregivers using MCDI prior to vocabulary training) and infants' preference for the target on vocabulary test trials (*r*=-.08, *p*=.56). This suggests that infants' comprehension *prior* to vocabulary training did not affect their looking behavior on the vocabulary test *after* the training.

**Outcome assessment (after vocabulary training).** A GLMM fitted to the mean %LT to target evaluated the effects of Condition, Age, and Age-by-Condition interaction. Significant effects included the effect of the intercept, $\beta$=0.17, *SE*=0.04, *t*=-3.95, *p*=.004, Age, $\beta$=-0.003, *SE*=0.001, *t*=-2.45, *p*=.016, and Age-by-Condition interaction (compared to the No Priming condition), $\beta$=-0.004, *SE*=0.002, *t*=2.20, *p*=.029. As with the baseline assessment, the distribution of residuals significantly deviated from normal (*p*<.05) and was left-skewed, rendering significance levels uninterpretable. We therefore applied an exponential transformation and re-ran the analyses. The results (see Table 4) did not substantially differ from the original model, except that the residuals were now normally distributed.

**Fig 5. The percent of infants' looking to the target on vocabulary test trials, averaged across two windows of analysis (367-2000 ms from the target word onset), corresponding to the two mentions of the target word (1000 ms and 3000 ms after image onset).** Asterisks represent significant t-values obtained from one-sample t-tests.

Two additional models—using the No Priming and the Switch Word as reference levels for Condition—revealed no significant difference between the No Priming and Switch Word conditions, β=-0.006, SE=0.05, t=-0.12, p=.91. The effects of the intercept were significant in both models, indicating that in both control conditions, infants preferred the target object (Fig 6). This outcome is consistent with the possibility that intrinsic preferences may have contributed to 12-month-olds' performance.

An additional set of models testing the effects of infants' comprehension of the words used in priming (measured as the mean %LT to the target on the vocabulary test, grouped by semantic neighborhood) after vocabulary training on their looking to the target at the

**Table 4. Results of mixed-effect model fitting with Condition as a fixed factor and Test Item and Participant as random factors for the intercept. Outcome assessment.**

|  | β | SE | T-value | p-value |
|---|---|---|---|---|
| (Intercept) | 0.52 | 0.04 | 10.20 | <.001* |
| Condition=No Priming | -0.05 | 0.05 | -0.91 | .37 |
| Condition=Switch Word | -0.05 | 0.04 | -1.07 | .27 |
| Age in days at Outcome, centered at median | 0.004 | 0.002 | -2.22 | .03* |
| Condition=No Priming*Age in days at Baseline | 0.004 | 0.002 | 1.96 | .05 |
| Condition=Switch Word*Age in days at Baseline | 0.003 | 0.002 | 1.36 | .18 |

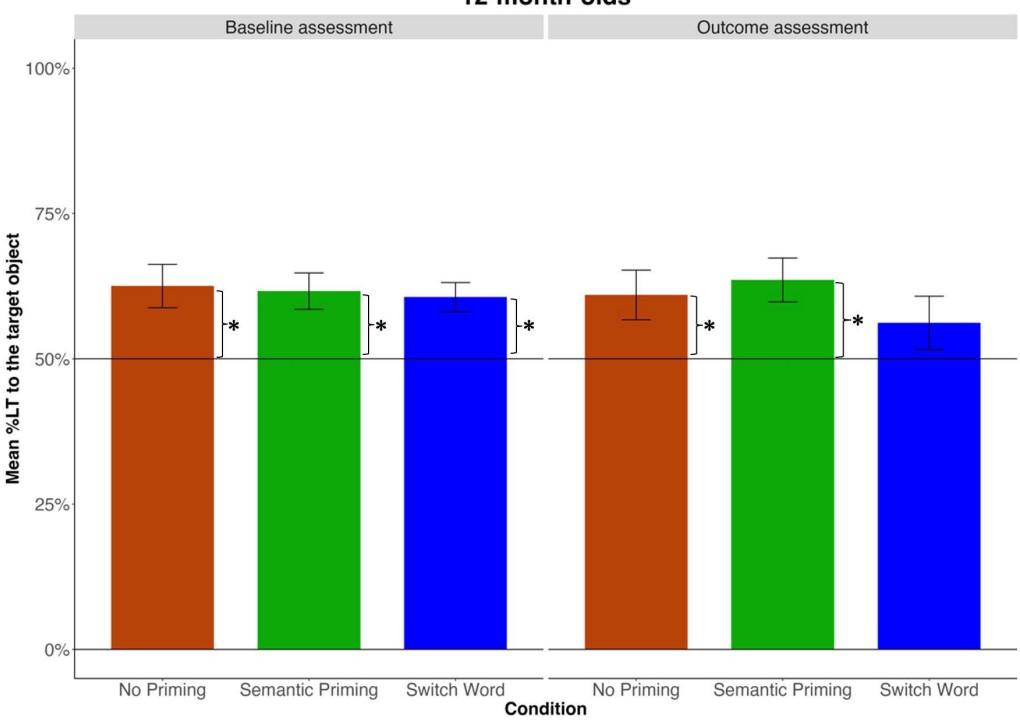

**Fig 6. Mean %LT to the target across two windows of analysis (after each mention of the target word) at baseline and outcome assessments, by Condition.** Asterisks represent significant effects of the intercept in a GLMM with the condition of interest entered as the reference level.

outcome assessment revealed no significant effects in either condition. This suggests that infants' performance on the baseline assessment was not due to limitations in infants' vocabulary before the vocabulary training.

**2. Time course.** Cluster-based permutation analysis identified no significant clusters of divergence among looking trajectories across conditions, before or after vocabulary training (see Fig 7). In fact, the timecourse data in all conditions mirrored one another, including the increase in looking to the target at the onset of image display, peaking around 800 ms. This pattern was evident at both baseline and outcome assessments, and in iCatcher and human coders alike (see Appendix 4).

The lack of differences among conditions after vocabulary training is counter to our predictions. We suspect that this may be due our choice of target and distractor objects. Target objects were chosen from the semantic neighborhoods that, on average, contain most words known to infants at this age, which may also suggest that infants are most familiar with the objects in those neighborhoods. Consistent with this possibility, infants' initial looking preference during test trials was for objects from those semantic neighborhoods, preferring images of fruits, animals, clothing, and vehicles in all conditions (Fig 8). This pattern was evident in both baseline and outcome assessments (Fig 7).

These results suggest that 12-month-olds' initial looking to the target test object on the experimental trials was not necessarily driven by verbal prompts, but rather by their preferences for images from familiar semantic neighborhoods. If this is the case, then their

**Fig 7. Time course of infants' looking during test trials at baseline and outcome assessments, by Condition.** Windows of analysis are shaded grey.

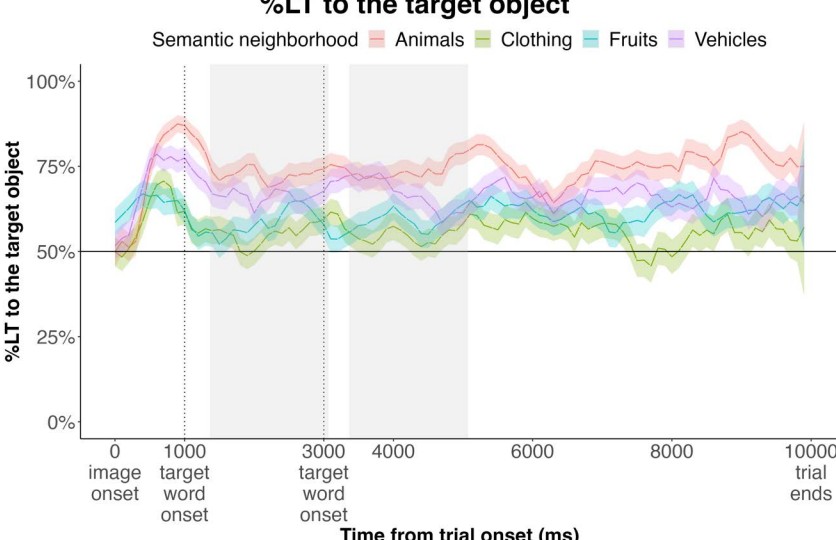

**Fig 8. Time course of infants' looking during vocabulary trials, grouped by semantic neighborhood.** Windows of analysis are shaded grey.

responses to the second prompt may provide a more sensitive test. To test this possibility, we conducted additional analyses, focusing this time on the window *after the second mention* of the target word. We reasoned that at this point, infants' early preferences had been resolved (their looking reverted to chance by 1700 ms, see Fig 7). At both baseline and outcome assessments, infants' looking during this window was above chance in the Semantic Priming condition, but not in the control conditions (Fig 9).

We also repeated the regression analyses for the second window only. These revealed no significant effects of Condition, Age, or Age-by-Condition interaction. This outcome suggests that 12-month-olds did not identify the referent of the novel noun in any condition.

**Discussion.** Twelve-month-olds' performance on the current task did not change even after an extended period of training on the names of the objects they would see during Priming. These infants continued to offer no evidence of establishing a representation for the meaning of a novel word during Priming and then using that representation to identify a candidate referent when one became visible. Neither infants' vocabulary prior to vocabulary training (infants' MCDI scores, reported by parents) nor infants' performance on vocabulary test trials was related to their performance on the main experimental procedure. Moreover, despite vocabulary training, 12-month-olds appeared to remain most familiar with the four of the semantic neighborhoods we selected for priming, which influenced their looking in all conditions. Alternatively, rather than looking to objects drawn from the more familiar semantic neighborhoods, infants' preferences may have been due to other factors, such as size, whether the object is graspable, or animacy. This possibility will be tested in future research.

## General discussion

In this investigation, we explored the developmental origins of the human ability to learn new information about things that are not perceptually available. Specifically, we examined 12- and 15-month-old infants' ability to establish a representation for the meaning of a novel word (however sparse), with no referent in sight, and use that representation to identify a candidate referent when one becomes visible. Previous literature had documented that infants as young

**Fig 9. Mean %LT to the target during the second window of analysis (3117–4750 ms) at baseline and outcome assessments, by Condition.** Asterisks represent significant effects of the intercept in a GLMM with the condition of interest entered as the reference level.

as 12 months can comprehend verbal reference to previously seen and named but currently hidden objects, but that it is not until 19 months that infants successfully infer meaning of a novel noun without any referent object in sight.

To the best of our knowledge, we provide the first evidence that infants as young as 15 months can represent something about the meaning of novel nouns in the absence of visible referents and draw on such representations when a candidate referent becomes available. Fifteen-month-olds in our experiment preferred to look to the object from the primed semantic neighborhood upon hearing the novel noun introduced during Priming but showed no such preference when the noun presented at Test differed from the one introduced during Priming. What is the best account for this pattern of findings? What did infants represent about the novel noun that allowed them to identify its referent when it became visible?

One possibility is that infants in the Semantic Priming condition (but in neither of the two control conditions) established a representation *of the hidden object*, realized that it was being named, and linked that representation to the novel noun. Prior research shows that by 15–16 months, infants can use familiar, known words to access representations of their referents, to interpret requests for hidden objects, even when such requests are referentially ambiguous, to update recently seen scenes, and to individuate hidden objects that they have never seen. Considered in the context of this evidence, it is indeed possible that 15-month-olds in the Semantic priming condition established a placeholder representation of the hidden object (however sparse) mapped it to the novel noun, and used this mapping to identify the referent of the novel noun when it became visible at Test. In contrast, infants in the No Priming condition did not have the requisite information about the hidden object to establish such

a representation and therefore could not identify the referent object at Test. Infants in the Switch Word condition did not hear the word used to label the hidden object; upon hearing an entirely new word, they could not identify either object as its referent.

Another possibility is that infants in the Semantic Priming condition *did not establish a representation of the hidden objects, per se.* Perhaps, instead, the activation of a semantic neighborhood was sufficient. If this were the case, the activated semantic field (or specific features of recently shown objects) guided their looking at test. In this case infants in the Switch Word condition should favor looking at the same object as those in the Semantic Priming condition. The evidence from the Switch Word condition reveals that this was not the case: the semantic neighborhood did not, on its own, yield the same effect as in the Semantic Priming condition. Alternatively, perhaps the entirely new novel noun in the Switch Word condition disrupted the activation of a semantic field or violated infants' expectations for the continuity of discourse. These possibilities warrant investigation to clarify the interpretation of our results.

Our findings also raise new questions for future research. First, it will be important to specify why 12-month-olds did not behave like 15-month-olds. One possibility is that the processing costs of simultaneously representing an unseen object and learning a new word are too steep for 12-month-olds, even after a period of vocabulary training. Alternatively, perhaps infants' ability to establish representations of a novel word's meaning, when no referent is visible in the naming episode, emerges later—between 12 and 15 months [54]. These possibilities will be explored in future investigations. In addition, it will be important to specify the content and breadth of 15-month-olds' representations of the novel words' meanings. This issue is currently under investigation.

This work also opens new avenues of investigation. First, we can now ask how 15-month-olds' ability to learn novel names for unseen objects influences their broader ability to learn new information from language. At issue is whether individual differences in infants' performance the current task predict their later learning from verbal testimony. We are currently testing this possibility in a longitudinal investigation that traces whether infants' performance in the Semantic Priming condition at 15 months predicts their success in learning new facts about unseen creatures at 24 months. It is also an open question whether infants' ability to link novel words with representations of the unseen is an entry point into naming and representing unseen or abstract phenomena more broadly, including our ability to map words onto concepts that have no visible referents (e.g., abstract concepts like 'justice', 'belief', 'square root of negative one'). Examining these possibilities is an essential step towards a better understanding how language enables learning, how it aids abstract reasoning, and how it helps us communicate about hypothetical scenarios, unobservable mental states, or imagined worlds.

## Supporting information

**S1 Appendix. Augmented MacArthur Short Form Vocabulary Checklist: Level II.**
(DOCX)

**S2 Appendix. Lookit demographic survey.**
(DOCX)

**S3 Appendix. Visual\* and auditory stimuli in Experiments 1 and 2.** *Source of images in depicted here: https://unsplash.com. Actual stimuli may differ (all stimuli are available on OSF).
(DOCX)

**S4 Appendix. A comparison of looking data coded by human coders and iCatcher+.**
(DOCX)

**S5 Appendix. A representative example of a Lookit code for Experiments 1 and 2, Switch Word condition.** Semantic Priming and No Priming codes look identical, except for the video recordings shown within the frame named "test trials". A single video recording contains both Familiarization and Test.
(DOCX)

## Acknowledgments

We thank Caitlin Draper, Courtney Goldenberg, Rachel Kritzig, Ren Mondesir, Kiki Ogbuefi, Mary Okematti, Katelyn Pass, Judith Roeder, Annalisa Romanenko, Murielle Standley, and Victoria Vizzini for help with pilot data collection. We thank Emily Apadula, Emily Yang, Matthew Bales, Matthew McGrory, Jacqueline Cantu, Cecilia Nam, Felicia Mou, Melany Morales, Megan Gonzales, and Leah Simon for their help with data collection and coding. We also thank Miriam Novack and Alexander LaTourrette for their contribution to stimuli creation, helpful discussion about the study design, and help with pilot data analyses.

## Author contributions

**Conceptualization:** Elena Luchkina, Sandra Waxman.

**Data curation:** Elena Luchkina.

**Formal analysis:** Elena Luchkina.

**Funding acquisition:** Elena Luchkina, Sandra Waxman.

**Investigation:** Elena Luchkina, Sandra Waxman.

**Methodology:** Elena Luchkina, Sandra Waxman.

**Project administration:** Elena Luchkina, Sandra Waxman.

**Resources:** Elena Luchkina, Sandra Waxman.

**Supervision:** Elena Luchkina, Sandra Waxman.

**Validation:** Elena Luchkina.

**Visualization:** Elena Luchkina.

**Writing – original draft:** Elena Luchkina, Sandra Waxman.

**Writing – review & editing:** Elena Luchkina, Sandra Waxman.

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
