## [Decision Letter · Decision Letter 0]

26 Jun 2024

PONE-D-24-17859Fifteen-month-olds represent never-seen objects and learn their namesPLOS ONE

Dear Dr. Luchkina,

Thank you for submitting your manuscript to PLOS ONE. After careful consideration, we feel that it has merit but does not fully meet PLOS ONE’s publication criteria as it currently stands. Therefore, we invite you to submit a revised version of the manuscript that addresses the points raised during the review process.

Although this paper is interesting and has the potential to contribute to the literature on cognitive and language development, there are some issues that need to be addressed before it can be accepted for publication. As stated by R1, the claim that 15 month old infants can form a mental representation and learn the name of never before seen objects is not strongly justified by the data--a more accurate claim would be that they can do so for an unseen or hidden object. Moreover, both reviewers highlight relevant research that should be used to contextualize this work. Finally, Expt. 2 may not be worth reporting given the high exclusion rate, as pointed out by R2. Provided that the authors are willing to address these issues and others raised in the reviews, the paper will be sent to the reviewers for re-evaluation and a decision will be rendered taking their input into account.

We look forward to receiving your revised manuscript.

Kind regards,

Laura Morett

Academic Editor

PLOS ONE

2. Thank you for stating the following financial disclosure: "NICHD NRSA Postdoctoral Fellowship to Elena Luchkina: GRANT13251277"  

3. Please expand the acronym “NICHD” (as indicated in your financial disclosure) so that it states the name of your funders in full.

Reviewers' comments:

Reviewer's Responses to Questions

**Comments to the Author**

1. Does the manuscript adhere to the experimental procedures and analyses described in the Registered Report Protocol?

If the manuscript reports any deviations from the planned experimental procedures and analyses, those must be reasonable and adequately justified.

Reviewer #1: Yes

Reviewer #2: Yes

2. If the manuscript reports exploratory analyses or experimental procedures not outlined in the original Registered Report Protocol, are these reasonable, justified and methodologically sound?

A Registered Report may include valid exploratory analyses not previously outlined in the Registered Report Protocol, as long as they are described as such.

Reviewer #1: Partly

Reviewer #2: Yes

3. Are the conclusions supported by the data and do they address the research question presented in the Registered Report Protocol?

The manuscript must describe a technically sound piece of scientific research with data that supports the conclusions. The conclusions must be drawn appropriately based on the research question(s) outlined in the Registered Report Protocol and on the data presented.

Reviewer #1: Yes

Reviewer #2: Partly

4. Have the authors made all data underlying the findings in their manuscript fully available?

Reviewer #1: Yes

Reviewer #2: Yes

5. Is the manuscript presented in an intelligible fashion and written in standard English?

Reviewer #1: Yes

Reviewer #2: Yes

6. Review Comments to the Author

Please use the space provided to explain your answers to the questions above. (Please upload your review as an attachment if it exceeds 20,000 characters)

Reviewer #1: The paper assesses infants' preference for looking at objects that have been semantically primed before. I did not review the Stage 1 report, so I will focus on the results and discussion for my review.

328: There is some newer work that is more positive: Pomiechowska, B., Bródy, G., Csibra, G., & Gliga, T. (2021). Twelve-month-olds disambiguate new words using mutual-exclusivity inferences. Cognition, 213, 104691.

Deviations from the registered protocol seem reasonable. The most significant deviation is the change of the analysis time window. Would the results be different if the original window had been chosen? Why did Ferguson use a longer window? Why is the “conventional” window the way it is? I think it would be good to be more explicit here.

Methods: How was the study implemented? Is the code available somewhere?

Analysis: Why is there no random effect for subjects? Some, if not most, individuals have more than one trial.

In line 288, you give 1.84 as the t-value threshold; in 304, it’s 1.69. Why are the values different? Which one was actually used?

307f: I would include the non-significant demographics in the final analysis nevertheless. Or what is the reason not to do so?

I’m not sure Experiment 2 is meaningful, given that you excluded twice as many infants as you included in the sample. It seems the procedure is too challenging for them.

502: Please report the correlations even if they are not significant.

506: How was vocabulary knowledge computed? Was the data aggregated by infant or by infant and category?

Discussion: I do not understand why children should look at the target in the no-priming condition (what they do). Any explanation? This result suggests something was off with the stimuli, the procedure, or the coding.

574: Such a design critically depends on the reliability of measurement. Did you assess something like split-half reliability? (I’d recommend the splithalfr package to account for the small number of trials). This is just a comment; no need to address it in this paper.

Reviewer #2: My answer to question 3 is "partly" because, in my opinion, the results are slightly overinterpreted. This can be easily fixed by phrasing the conclusions more carefully. I detailed my suggestions in the attached file.

7. PLOS authors have the option to publish the peer review history of their article (what does this mean? ). If published, this will include your full peer review and any attached files.

**Do you want your identity to be public for this peer review?** For information about this choice, including consent withdrawal, please see our Privacy Policy .

Reviewer #1: No

Reviewer #2: No

---

## [Author Response · Author response to Decision Letter 0]

17 Sep 2024

Dear Dr. Lorett,

Please consider our revised manuscript PONE-D-24-17859 for publication in PLOS One. We are grateful to you for your and reviewers for such thoughtful feedback and for offering us the opportunity for revision. In the attached pdf document, we provide a point-by-point response to each reviewer’s comments. We attach the revised manuscript and a track-changes copy.

We hope that you will agree that by responding to this very constructive set of reviews, we have now produced a manuscript that is acceptable for publication in PLOS One.

Sincerely,

Authors

---

## [Decision Letter · Decision Letter 1]

8 Oct 2024

PONE-D-24-17859R1Fifteen-month-olds can learn names of unseen objects in their absencePLOS ONE

Dear Dr. Luchkina,

Thank you for submitting your manuscript to PLOS ONE. After careful consideration, we feel that it has merit but does not fully meet PLOS ONE’s publication criteria as it currently stands. Therefore, we invite you to submit a revised version of the manuscript that addresses the points raised during the review process.

Although the authors were responsive to the previous set of comments, there are some additional aspects of this manuscript that still need to be addressed, including the analyses employed and interpretation of the results.  If the authors are willing to address these remaining issues, the manuscript will be re-assessed for suitability for publication in PLOS One.

We look forward to receiving your revised manuscript.

Kind regards,

Laura Morett

Academic Editor

PLOS ONE

Reviewers' comments:

Reviewer's Responses to Questions

**Comments to the Author**

1. Does the manuscript adhere to the experimental procedures and analyses described in the Registered Report Protocol?

If the manuscript reports any deviations from the planned experimental procedures and analyses, those must be reasonable and adequately justified.

Reviewer #1: Yes

Reviewer #2: Yes

2. If the manuscript reports exploratory analyses or experimental procedures not outlined in the original Registered Report Protocol, are these reasonable, justified and methodologically sound?

A Registered Report may include valid exploratory analyses not previously outlined in the Registered Report Protocol, as long as they are described as such.

Reviewer #1: Yes

Reviewer #2: Yes

3. Are the conclusions supported by the data and do they address the research question presented in the Registered Report Protocol?

The manuscript must describe a technically sound piece of scientific research with data that supports the conclusions. The conclusions must be drawn appropriately based on the research question(s) outlined in the Registered Report Protocol and on the data presented.

Reviewer #1: Yes

Reviewer #2: Partly

4. Have the authors made all data underlying the findings in their manuscript fully available?

Reviewer #1: Yes

Reviewer #2: Yes

5. Is the manuscript presented in an intelligible fashion and written in standard English?

Reviewer #1: Yes

Reviewer #2: Yes

6. Review Comments to the Author

Please use the space provided to explain your answers to the questions above. (Please upload your review as an attachment if it exceeds 20,000 characters)

Reviewer #1: I thank the authors for attending to my comments. They have alleviated all my concerns but one.

It is still unclear why there is no random effect for subject in the analysis for %LT. In the manuscript, they say that children received ideally 4 trials (2.9 on average) within condition. Thus, there are multiple data points for each subject which should be accounted for by a random effect. It does not logically follow from the between subject design for condition that there is no variation *within* condition. This would only be the case if the data was aggregated for each individual within condition but as I read line 375, there was more than one trial per subject in the analysis. Without the random effect, there is no differentiation between trials coming from the same individual and trials coming from different individuals.

Reviewer #2: I chose "partly" as my response to question 3 because I disagree with the authors' strong interpretation of the results. See details in comment 1 in the review.

My answer to question 4 is "yes", however I did not see the 5 tables with different data and stat summaries that are referenced in the text.

7. PLOS authors have the option to publish the peer review history of their article (what does this mean? ). If published, this will include your full peer review and any attached files.

**Do you want your identity to be public for this peer review?** For information about this choice, including consent withdrawal, please see our Privacy Policy .

Reviewer #1: No

Reviewer #2: No

---

## [Author Response · Author response to Decision Letter 1]

11 Dec 2024

Dear Dr. Morett,

Thank you again. We are grateful to you and the reviewers for your thoughtful feedback on our manuscript, PONE-D-24-17859R1. In the revision, we have addressed reviewers’ remaining concerns and suggestions. Below we provide a point-by-point response to each reviewer’s comments. We hope that you will agree that by responding to this very constructive set of reviews, we have now produced a manuscript that is acceptable for publication in PLOS One.

Sincerely,

Authors

Please note that the line numbers correspond to those in the clean version (not with track changes).

Reviewer 1

This reviewer had only one remaining concern, one that we have attended to carefully. The reviewer noted that it was unclear why our analyses included no random effect for subject in the analysis for %LT.

We thank R1 for careful attention to our analyses. We agree that in principle, including a grouping variable at the level of randomization would be ideal. Yet here, when we add Participant as another grouping (random) variable in addition to Trial1, only a single data point is available to estimate the intercept within each group, and the variance of the intercept is 0. This results in a singular fit error. Even when we remove Trial as a grouping variable, the variance of the intercept due to Participant remains negligible (close to 0), and the model again returns a singular fit error. Moreover, adding or removing grouping variables does not substantially affect our coefficients or their significance (see Table 1 below). We have therefore opted to report the model with Trial (and not Participant) as a grouping variable in manuscript. If you prefer that we include Table 1 as an appendix in the manuscript, we would be happy to do so.

Model Condition Beta SE T-value p-value

Mean %LT ~ Condition+ (1|Trial) + (1|Participant) Intercept 0.1784 0.0446 4.00 .0032

Mean %LT ~ Condition+ (1|Trial) + (1|Participant) No Priming -0.1784 0.0530 -3.36 .0010

Mean %LT ~ Condition+ (1|Trial) + (1|Participant) Switch Word -0.1169 0.0468 -2.50 .0138

Mean %LT ~ Condition+ (1|Trial) Intercept 0.1784 0.0446 4.00 .0032

Mean %LT ~ Condition+ (1|Trial) No Priming -0.1784 0.0530 -3.36 .0010

Mean %LT ~ Condition+ (1|Trial) Switch Word -0.1169 0.0468 -2.50 .0138

Mean %LT ~ Condition+ (1|Participant) Intercept 0.1817 0.0359 5.06 .0000013

Mean %LT ~ Condition+ (1|Participant) No Priming -0.1888 0.0539 -3.50 .00063

Mean %LT ~ Condition+ (1|Participant) Switch Word -0.1194 0.0478 -2.50 .01367

Mean %LT ~ Condition Intercept 0.1817 0.0359 5.06 .0000013

Mean %LT ~ Condition No Priming -0.1888 0.0539 -3.50 .00063

Mean %LT ~ Condition Switch Word -0.1194 0.0478 -2.50 .01367

Table 1. Coefficients from a GLMM predicting Mean% LT in 15-month-olds

1 Trial is referred to as ‘Test Item’ in the text of the manuscript. Each trial probed a word-image mapping—a test item—from a different semantic neighborhood.

Reviewer 2

This reviewer found the manuscript much improved but mentioned a few remaining questions and suggestions.

We thank R2 for paying such a close attention to the consistency of our argument and the accuracy of our text. This feedback has been instrumental.

“I recommend softening or rephrasing the strong assertion about infants’ understanding of the referential nature of words…”

We thank R2 for this suggestion. Following the suggestion, we removed the paragraph about reference from the introduction. We have also softened our assertion throughout.

“I am not entirely convinced that the design and the results of the current work provide a direct test of infants’ ability to map words to mental representations or to category of objects…”

Our design permits us to assess infants’ expectation for the (formerly hidden) referent of the novel word when it becomes visible at Test. As in most analyses based on a word-mapping task at Test, it does not permit us to test the nature or the breadth of infants’ representation of the novel word’s referent. It does, however, provide insight into infants’ ability to map a novel noun to an object unseen prior to the Test phase on the basis of information gleaned earlier (Priming phase) in the absence of any visible referent for that word, and to then to recruit this mapping later at Test. Moreover, we now note (in the GD; lines 691-693) that we are currently testing the breadth of infants’ interpretation of the novel word. Once those data are available, we will have more evidence regarding infants’ ability to link words with kinds, or categories, of unseen objects.

“In the Semantic Priming condition … infants could have inferred the meaning of the novel label during the test … Moreover, there is no direct assessment of what kind of meaning infants attach to the label of the unseen object while it is still unseen. … Although the task is very interesting, it does not provide direct evidence that infants map the novel word onto their representation of the hidden object or that they need this phase of the experiment at all to identify the target.”

We agree with R2’s point that our task provides no direct assessment of the kind of representation that infants attach to the novel label of the unseen object during the Priming phase. Instead, we probe infants’ expectations in each condition at Test. We also agree that on the basis of the information provided in the Priming phase, infants could not know which particular member of a semantic neighborhood might be the referent of the novel word until they see the test images. Seeing those objects may indeed have influenced infants’ knowledge by clarifying the mapping.

Nevertheless, to succeed in our task (as they did), infants must have created some representation, however sparse, that supported their preference for the target object (and not the foil) when it appeared, de novo, at Test. Importantly, the evidence we report here is consistent with the prediction: Infants’ performance at test varies significantly as a function of the information provided during Priming. We now discuss this in the GD (lines 665-676).

“The comparison between the Semantic Priming and the Switch Word conditions sheds light to an extent on how the absent refence phase and the test work together for infants’ ability to identify the target. We do see that infants need a consistent reference across the absent reference phase and the test to succeed … .”

We are happy to hear that we have now reported the results in such a way as to clarify this key finding.

“It is possible that infants failed in the Switch Word condition because they were confused by hearing two labels. They first were invited to “play a game and find a wug” and then after the delay they suddenly had to find the “dawnoo” which could have confused them”

We now address this possibility in the GD (lines 677-685): “We think there is no reason for such a confusion, unless infants expected continuity of discourse within a trial. If infants expected a correspondence between the pragmatics of the situation—the actor first ostensibly labels a non-visible object and then invites the infant to play a finding game—and the words the actor used, then infants must had realized that the actor was using the novel noun to refer to the hidden object and expected her to continue using that word. This explanation is also consistent with our interpretation that infants formed some representation of the hidden object and linked it the novel word.”

“You might argue that in the Semantic Priming condition infants start looking at the target after the first naming event that occurs before the objects actually become visible, so they mapped the word on a mental image of an unseen object first. However, it is possible that the word was still present in infants’ memory when they saw the objects because it was the last one mentioned and with a minimum delay.”

Here, we offer a 2-part response to R2’s point.

First, this would have been a concern if the word-object (i.e. image of the object) mapping was learned in the presence of the object. In that case, retaining the word in the short-term memory may have been sufficient to access a mapping between that word and the object and prompt infants to look to the target object at Test. However, because infants had not seen the target object until it appeared at Test, recalling the word could not have called up a memory of any particular object. Instead, the results indicate that infants connected the word to a more abstract representation.

Second, notice that infants’ looking to the target in the Semantic Priming condition increased after the second mention of the word, paralleling the increase after the first mention (see Figure 3). This provides additional evidence that infants’ looking reflects a word-referent mapping (rather than a carryover activation of a specific semantic field).

R2 also suggested we revise the title once more to better reflect our findings.

We have done so. The revised title is “Fifteen-month-olds map novel words to referents that are absent during naming.” We believe that this title accurately reflects our findings.

R2 suggested that the abstract can be formulated in a more clear and concise way.

We thank R2 for catching these errors and making helpful suggestions about rewording the abstract. We have implemented the suggested changes.

“Line 151, there is a different format of citation here (Swingley & Aslin, 2000) and it is not included in the references.”

We have now made the citation format consistent with the rest of the manuscript and included this citation to the reference list.

R2 asked that we clarify how many participants were excluded from the final sample.

We thank R2 for this suggestion. We now include an explicit statement that the 31 additional participants were excluded from the final analyses (lines 185-188).

R2 suggested that we remove ‘novel’ objects and use ‘unfamiliar’ objects instead to refer to the items we present at Test because there is a chance that infants saw such objects in the past.

We have implemented both suggested changes throughout the manuscript.

R2 suggested that we do not use the word proportion in reference to our DV %Mean LT.

We have changed ‘proportion’ to ‘percent’ throughout.

R2 raised a question whether the use of Holm-Bonferroni adjustment in a regression model with multiple predictors was justified.

The use of the adjustment just depends on how you view the hypothesis. It’s either two hypotheses tested in one model or one hypothesis with two coefficients. We originally applied the Holm-Bonferroni adjustment to be conservative and cover both of those interpretations of the hypothesis. In this case, given that the effects of interest are significant with or without the adjustment, we have removed it based on R2’s comment (line 628).

“Lines 391-393. Please provide the beta coefficient for the intercept with all the accompanying statistics, like you do for the other two comparisons.”

Done (lines 382-383)

R2 suggested that we perform comparisons to the chance level by testing the significance of the intercept in models (with each of our conditions treated as the reference level in a separate model) instead of using t-tests.

We did so for both experiments (lines 387-393, 547-553, and 582-587).

R2 asked us to explain why we did not test the knowledge of distractor object names in Exp. 2.

We now provide a statement that we did not test infants’ knowledge of the distractor objects’ names because this knowledge was not relevant to 12-month-olds ability to benefit from semantic priming (lines 491-493).

R2 asked us to clarify why the slope of age in days in Exp. 2 was negative.

We now added a statement that the observed effect of age is likely a Type 1 error (line 527). In preliminary analyses, the effects were entered into the same model, which we now state explicitly (lines 375 and 525).

“I could not find tables 1-5. They are being referenced in the text and there are captions for them under “Supporting information captions” but I did not find them in the downloadable Appendices, in the text, or on OSF.”

We apologize for this. We will be sure to check if the tables are available in the PDF version of our submission and add them on OSF if necessary.

R2 also pointed out a number of spelling, punctuation and style errors and suggested better wording in a few places.

We are grateful to R2 for their careful attention to detail in our manuscript. We have corrected the errors and improved the wording in every case.

---

## [Decision Letter · Decision Letter 2]

8 Jan 2025

PONE-D-24-17859R2Fifteen-month-olds map novel words to referents that are absent during namingPLOS ONE

Dear Dr. Luchkina,

Thank you for submitting your manuscript to PLOS ONE. After careful consideration, we feel that it has merit but does not fully meet PLOS ONE’s publication criteria as it currently stands. Therefore, we invite you to submit a revised version of the manuscript that addresses the points raised during the review process.

I thank the authors for their attention to the reviewers' comments. There are a few minor points that remain to be addressed, including inclusion of random intercepts by participant (R1), discussion of alternative explanations for the findings (R2), and proofreading for remaining language issues (R2). If the authors can address these remaining points, I will review their responses and render a decision without sending the manuscript out for another round of review.

We look forward to receiving your revised manuscript.

Kind regards,

Laura Morett

Academic Editor

PLOS ONE

Journal Requirements:

Reviewers' comments:

Reviewer's Responses to Questions

**Comments to the Author**

1. Does the manuscript adhere to the experimental procedures and analyses described in the Registered Report Protocol?

If the manuscript reports any deviations from the planned experimental procedures and analyses, those must be reasonable and adequately justified.

Reviewer #1: Yes

Reviewer #2: Yes

2. If the manuscript reports exploratory analyses or experimental procedures not outlined in the original Registered Report Protocol, are these reasonable, justified and methodologically sound?

A Registered Report may include valid exploratory analyses not previously outlined in the Registered Report Protocol, as long as they are described as such.

Reviewer #1: Yes

Reviewer #2: Yes

3. Are the conclusions supported by the data and do they address the research question presented in the Registered Report Protocol?

The manuscript must describe a technically sound piece of scientific research with data that supports the conclusions. The conclusions must be drawn appropriately based on the research question(s) outlined in the Registered Report Protocol and on the data presented.

Reviewer #1: Yes

Reviewer #2: Partly

4. Have the authors made all data underlying the findings in their manuscript fully available?

Reviewer #1: Yes

Reviewer #2: Yes

5. Is the manuscript presented in an intelligible fashion and written in standard English?

Reviewer #1: Yes

Reviewer #2: Yes

6. Review Comments to the Author

Please use the space provided to explain your answers to the questions above. (Please upload your review as an attachment if it exceeds 20,000 characters)

Reviewer #1: The authors have addressed all my concerns.

One last comment: I think it makes conceptually more sense to include random intercepts for participants instead of trials (because data points coming from the same participant are likely to be more similar than data points coming from the same trial).

Reviewer #2: In this last (hopefully) revision of the manuscript I ask that the authors dedicate more attention to the remaining alternative explanations of the results and present a more thorough argumentation to support their default interpretation. Minor language errors need some attention as well.

7. PLOS authors have the option to publish the peer review history of their article (what does this mean? ). If published, this will include your full peer review and any attached files.

**Do you want your identity to be public for this peer review?** For information about this choice, including consent withdrawal, please see our Privacy Policy .

Reviewer #1: No

Reviewer #2: No

---

## [Author Response · Author response to Decision Letter 2]

7 Mar 2025

Dear Dr. Morett,

Thank you again. We are grateful to you and the reviewers for your thoughtful feedback on our manuscript, PONE-D-24-17859R2. In the revision, we have addressed reviewers’ remaining concerns and suggestions. Below we provide a point-by-point response to each reviewer’s comments. We hope that you will agree that by responding to this very constructive set of reviews, we have now produced a manuscript that is acceptable for publication in PLOS One.

Sincerely,

Authors

Please note that the line numbers correspond to those in the clean version (not with track changes).

Reviewer 1

Reviewer 1 asked us to report the results of our regression analyses including Participant as a grouping factor for the intercept.

We followed R1’s advice and now report the results of our regression analyses in both experiments with Participant grouping factor included (lines 397-399, 547-549, and 581-583).

Reviewer 2

Reviewer 2 asked us to explicitly acknowledge alternative interpretations of our results and include more thorough and detailed argumentation supporting their default interpretations.

We have done so in the General Discussion (lines 682-691) of the revised manuscript. We now explicitly state that the interpretations suggested by R2 are indeed plausible and will be explored in future investigations.

Furthermore, we have edited the manuscript throughout to temper the original claim about infants’ representations of hidden referent and now emphasize that infants may represent something about the meaning of the novel noun, but not necessarily the hidden referent. Instead, we bring up the idea about infants’ representation of the hidden referent of the novel noun in the General Discussion as one of the plausible interpretations of our results. We then mention another alternative, one proposed by R2, as a plausible interpretation of our findings as well.

We also changed the title of the manuscript back to the original one, following R2’s advice.

We have also thoroughly proofread the manuscript and corrected all typos, word repetitions or omissions, and other errors.

We thank you and the reviewers for detailed feedback and advice.

---

## [Editor Report · Decision Letter 3]

11 Mar 2025

Semantic priming supports infants’ ability to learn names of unseen objects

PONE-D-24-17859R3

Dear Dr. Luchkina,

We’re pleased to inform you that your manuscript has been judged scientifically suitable for publication and will be formally accepted for publication once it meets all outstanding technical requirements.

Kind regards,

Laura Morett

Academic Editor

PLOS ONE

Additional Editor Comments (optional):

I thank the authors for addressing the remaining comments. Now that all comments have been adequately addressed, I am pleased to recommend this manuscript for publication in PLOS One.
---

## [Editor Report · Acceptance letter]

PONE-D-24-17859R3

PLOS ONE

Dear Dr. Luchkina,

I'm pleased to inform you that your manuscript has been deemed suitable for publication in PLOS ONE. Congratulations! Your manuscript is now being handed over to our production team.

Kind regards,

on behalf of

Dr. Laura Morett

Academic Editor

PLOS ONE